# Proceedings of the 7^th^ Asia Dengue Summit, June 2024

**DOI:** 10.3390/vaccines13050493

**Published:** 2025-05-02

**Authors:** Zulkifli Ismail, Duane J. Gubler, Tikki Pangestu, Usa Thisyakorn, Nattachai Srisawat, Daniel Goh, Maria Rosario Capeding, Lulu Bravo, Sutee Yoksan, Terapong Tantawichien, Sri Rezeki Hadinegoro, Kamran Rafiq, Eng Eong Ooi

**Affiliations:** 1Department of Pediatrics, KPJ Selangor Specialist Hospital, Shah Alam 40300, Malaysia; 2Program in Emerging Infectious Diseases, Duke-NUS Medical School, Singapore 169857, Singapore; 3Yong Loo Lin School of Medicine, National University of Singapore, Singapore 117597, Singapore; 4Executive Committee of Tropical Medicine Cluster, Chulalongkorn University, Bangkok 10330, Thailand; 5Faculty of Tropical Medicine, Mahidol University, Bangkok 73170, Thailand; 6Center of Excellence in Critical Care Nephrology, Faculty of Medicine, Chulalongkorn University, Bangkok 10330, Thailand; 7Excellence Center for Critical Care Nephrology, King Chulalongkorn Memorial Hospital, Bangkok 10330, Thailand; 8Division of Paediatric Pulmonary Medicine and Sleep, Khoo Teck Puat National University Children’s Medical Institute, National University Hospital, Singapore 119074, Singapore; 9Research Institute for Tropical Medicine, Muntinlupa City 1781, Manila Metro, Philippines; 10College of Medicine, University of the Philippines Manila, Manila 1000, Philippines; 11Center for Vaccine Development, Institute of Molecular Biosciences, Mahidol University, Bangkok 73170, Thailand; 12Division of Infectious Diseases, Department of Medicine, Chulalongkorn University, Bangkok 10330, Thailand; 13Department of Child Health, Faculty of Medicine, Universitas Indonesia, Jakarta 10430, Indonesia; 14International Society of Neglected Tropical Diseases, London WC2H 9JQ, UK

**Keywords:** dengue, vector control, vaccines, antigenic evolution, youth

## Abstract

**Background:** The 7^th^ Asia Dengue Summit (ADS), titled “Road Map to Zero Dengue Death”, was held in Malaysia from 5 to 7 June 2024. The summit was co-organized by Asia Dengue Voice and Action (ADVA); Global Dengue and *Aedes*-Transmitted Diseases Consortium (GDAC); Southeast Asian Ministers of Education Tropical Medicine and Public Health Network (SEAMEO TROPMED); Fondation Mérieux (FMx); and the International Society for Neglected Tropical Diseases (ISNTD). **Objectives:** Dengue experts from academia and research, as well as representatives from the Ministries of Health, Regional and Global World Health Organization (WHO), and International Vaccine Institute (IVI), came together to highlight the crucial need for an integrated approach for dengue control and achieve the target of zero dengue deaths. **Methods:** With more than 50 speakers and delegates from over 28 countries, twelve symposiums, and three full days, the 7^th^ ADS highlighted approaches to curb the growing danger of dengue. The summit included topics ranging from emerging dengue trends, insights from dengue human infection models, the immunology of dengue, and vaccine updates to antivirals and host-directed therapeutics. **Conclusions:** The 7^th^ Asia Dengue Summit reinforced the importance of an integrated, collaborative approach to dengue prevention and control. By bringing together diverse stakeholders and launching innovative initiatives such as the Dengue Slayers Challenge, the summit advanced the regional and global agenda to achieve zero dengue deaths. The exchange of knowledge and strategies at the summit is expected to contribute significantly to improved dengue management and community engagement in affected regions.

## 1. Introduction

The Asia Dengue Voice and Action (ADVA) is a scientific working group dedicated to dengue control across the region through collaboration with academia, industry, and non-government organizations. One of the core educational initiatives by the ADVA is the Asia Dengue Summit (ADS) held every year in collaboration with GDAC, SEAMEO TROPMED, FMx, and ISNTD. Following the inaugural conference held in 2016, every year, ADS focuses on key issues such as the growing global public health burden of dengue, outbreak prediction and surveillance, challenges in vaccination, travel-related dengue, and vector control.

The 7^th^ Asia Dengue Summit (7^th^ ADS) was held in Malaysia from 5 to 7 June 2024. With 50 speakers and 550 delegates from over 28 countries, thirteen symposiums, and three full days, the 7^th^ ADS highlighted approaches to curb the growing danger of dengue. The summit included topics ranging from emerging dengue trends, insights from dengue human infection models, the immunology of dengue, and vaccine updates to antivirals and host-directed therapeutics. Healthcare professionals, researchers, epidemiologists, and representatives from the Ministries of Health came together to highlight the crucial need for an integrated approach for dengue control to achieve the target of zero dengue deaths. In this special symposium, experts discussed dengue clinical scenarios, identified challenges, and shared clinical experiences to advance dengue case management. The inaugural ADVA Junior Achievement challenge highlighted the youth’s potential in developing creative and innovative solutions for dengue control. This report summarizes key highlights from the speaker presentations during the 7^th^ Asia Dengue Summit.

## 2. Wolbachia Success Stories

The World Mosquito Program’s (WMP) *Wolbachia* innovation is a safe, self-sustaining, one-time intervention operational across three continents and has shown promising results in reducing dengue incidence. The *Wolbachia* (*w*Mel) method works by introducing the naturally occurring bacteria, *Wolbachia*, into the *Aedes aegypti* mosquito population and rendering them incapable of transmitting diseases such as dengue, Zika, chikungunya, and yellow fever [1]. The World Health Organization (WHO) Vector Control Advisory Group (VCAG) has endorsed the public health value of the *Wolbachia* innovation against dengue [2].

The AWED trial (Applying *Wolbachia* to Eliminate Dengue) was the first cluster randomized controlled trial to evaluate the efficacy of the large-scale deployment of *Wolbachia*-infected *Aedes aegypti* mosquitoes in reducing the incidence of dengue at a single site in Yogyakarta, Indonesia [3]. The primary analysis of the AWED trial showed positive results with a 77% reduction in the incidence of virologically confirmed dengue (VCD) (67 VCDs amongst 2905 participants (2.3%)) vs. the untreated arm (318/3401 (9.4%)) (OR 0.23, 95% CI, 0.15 to 0.35; *p* = 0.004) and an 86% reduction in hospitalized VCD in *w*Mel-treated clusters compared to untreated areas (13 hospitalizations for VCD amongst 2905 participants (0.4%) vs. 102 hospitalizations for VCD amongst 3401 participants (3%) from the untreated arm) [4]. A secondary analysis of the AWED trial demonstrated an 83% reduction in the incidence of notified dengue hemorrhagic fever (DHF) during the fully treated versus untreated periods. Furthermore, there was an 83% reduction in the application of insecticide spraying in *w*Mel intervention areas, and a 39.6% reduction in the annual cost of insecticide spraying in Yogyakarta city [5].

Following the launch of the Wolbachia Malaysia project in 2017, *w*AlbB-carrying *Ae. Aegypti* were deployed in eleven dengue hotspots in the Klang Valley around Kuala Lumpur. Data collected from twenty high-rise residential areas with *Wolbachia* release demonstrated a 62.4% reduction in dengue fever incidence. So far, *Wolbachia* deployments have been carried out across forty localities in eight states in Malaysia, with the Malaysian Ministry of Health planning further coverage of dengue hotspots as a national rollout program [6].

## 3. Dengue Early Warning Tools

Strong vector surveillance systems are crucial to developing early warning systems to prompt intervention strategies to minimize public health impacts. Meteorological parameters and entomological parameters are used as predictors in dengue forecasting models [7].

DenMap, a web-based dengue surveillance system, is an excellent example of a real-time system for monitoring dengue outbreaks in Malaysia. The e-Notifikasi database (notified cases) and e-Dengue database (registered cases) in Malaysia enable the notification of dengue cases to the Ministry of Health within 24 h, facilitating the development of a rapid geocoding and visualization application. DenMap generates a quick, real-time visual display of notified and registered cases on Google Maps, providing an early warning of dengue hotspots, clusters, and spatial and temporal distributions of cases. DenMap is not only easy to use and flexible but is also a cost-effective solution to predict future dengue outbreaks [8].

The early warning and response system (EWARS), established in Mexico in 2012, underwent further development in 2014 to validate prediction algorithms using epidemiological alarm indicators (age and circulating serotype), entomological alarm indicators (positivity in laboratory tests, average of positive ovitraps, and average vector eggs per block), and meteorological alarm indicators (temperature, humidity, and rainfall). In 2018, Mexico incorporated EWARS into its national platform for integrated epidemiological surveillance.

Over 17 countries, including several in the Southeast Asia and Western Pacific regions, are either implementing or in the process of integrating EWARS into their national dengue control programs [9]. Recent evidence from the National Vector Control Program in Mexico validating the efficacy of EWARS highlights that adequate and timely responses to alarm signals play a crucial role in notably reducing dengue outbreaks and hospitalizations [10].

## 4. Insights from the Dengue Human Infection Models (DHIMs)

There are significant gaps in the current dengue countermeasure portfolio. As of now, there are no antivirals or monoclonal antibodies approved for the prevention or treatment of dengue. Particular challenges in the development of dengue countermeasures include the presence of four dengue serotypes, a lack of clarity on immunological and pathologic profiles, difficulty in measuring immune response, and the absence of accurate animal infection models that can replicate human infection. In order to overcome these challenges, a consortium of investigators was established with the objective of developing safe dengue human infection models (DHIMs). The main objective of DHIMs is to analyze the clinical, virologic, immunologic, and serologic features of mild dengue in a controlled and safe environment [11]. The key goals of DHIMs are to assess the potential clinical benefit of candidate vaccines and antiviral drugs to protect against infection with all four DENV serotypes, to validate host biomarkers of infection and protection, and to study host–virus interactions to understand the pathogenesis of the infection [12].

In a recent study, a phase 1, open-label assessment of the DENV-1 live-virus human challenge strain (DENV-1-LVHC strain 45AZ5) in twelve healthy adult volunteers resulted in an uncomplicated dengue illness that was well-tolerated without major safety signals. This first-in-human study of the DENV-1-LVHC strain indicates that it may be a potential DHIM candidate to test dengue countermeasures [13]. In another phase 1 open-label study in a low-dose dengue virus 3 human challenge model, the DENV-3 strain CH53489 resulted in clinical, laboratory, and immunological features consistent with mild-to-moderate dengue in all participants [14]. These DHIMs represent a unique opportunity to understand immune correlates of DENV infection risk and to assess the potential clinical benefit of candidate countermeasures.

## 5. Potential Biomarkers for Severe Dengue

Ongoing research to identify clinically usable biomarkers to predict severe dengue (SD) has shown promising results. Using an integrated multi-cohort analysis framework of seven gene expression datasets across five countries, Robinson et al. identified a twenty-gene set to predict SD [15]. This gene set is generalizable across ages, host genetic factors, and virus strains and has been validated in three retrospective datasets and one prospective cohort. Another iterative multi-cohort analysis of eleven public datasets from seven countries identified an eight-gene machine learning model that predicted SD progression (SDp) in an independent validation cohort [16].

Immune profiling of peripheral blood mononuclear cells by mass cytometry can also be used to delineate immune features associated with SDp. Early activation of innate immune response is a characteristic feature of dengue; on the other hand, SDp is characterized by the activation of IgG-secreting plasma cells and memory and regulatory T cells. Additionally, reduced human leukocyte antigen (HLA)–DR expression and increased CD64 expression on myeloid cells are hallmarks of SDp. The differences in innate immune response are exaggerated in children with higher proinflammatory NK cells, lower increases in CD16+ monocytes, and higher expressions of FcγR CD64 on myeloid cells. Furthermore, the immune cell composition varies across SD categories, with higher cDC1, cDC2, and Treg abundance in organ impairment and higher expansion of plasma cells in DHF/DSS [17].

In a recent study, Limothai et al. identified that circulating microRNA (miRNA) expression patterns in severe dengue patients, particularly miR-574-5p and miR-1246, can be used as a potential diagnostic and prognostic biomarker for SDp upon hospital admission [18]. Though several candidate biomarkers have been identified, they need further validation in larger prospective cohorts across all four dengue viruses, as well as age and host genetic factors. Additionally, such prognostic assays need to be reliable and affordable, and they should also have fast turnaround times and work in resource-limited settings.

## 6. Dengue Vaccine Update

Despite CYD-TDV (Dengvaxia™) receiving market authorization, clinical trials showed higher rates of hospitalized/severe dengue in baseline seronegative individuals with certain serotypes. Therefore, CYD-TDV requires pre-screening to determine serostatus before vaccination and is only recommended in baseline seropositive individuals who are 6 years and older [19].

Another tetravalent dengue vaccine, TAK-003, has been prequalified by the World Health Organization (WHO) and is recommended in children aged 6–16 years in settings with high dengue burden and transmission intensity. TAK-003 is now eligible for procurement by UN agencies, including the United Nations International Children’s Emergency Fund (UNICEF) and the Pan American Health Organization (PAHO) [20]. The first public program for TAK-003 vaccination in Brazil in the 6-to-16-year age group began in February 2024 [21]. In order to accelerate dengue vaccine access in endemic countries, plans are underway to manufacture TAK-003 multi-dose vials with the aim of improving manufacturing capacity by 100 million doses/year by 2030 [22].

The safety and efficacy of another vaccine candidate (Butantan–dengue vaccine) against VCD caused by DENV-1 and DENV-2, irrespective of baseline serostatus, was established in a phase 3, double-blind trial in Brazil during the 2-year follow-up period [23]. Further results from the 5-year follow-up period are anticipated. Another promising vaccine candidate is the modified mRNA vaccine encoding epitopes from dengue nonstructural protein, with a preliminary animal study showing a potent T cell response and protection against DENV infection [24]. Future studies are needed to elucidate the role of T cell-mediated immunity in the presence or absence of anti-DENV neutralizing antibodies.

## 7. Dengue Monoclonal Antibody Update

The DENV nonstructural protein 1 (NS1) plays an important role in dengue pathogenesis, especially during the critical phase, by mediating endothelial hyperpermeability, vascular leakage, thrombocytopenia, and hemorrhage, making it an attractive target for antiviral therapies. Preliminary animal studies have shown the efficacy of humanized anti-NS1 monoclonal antibodies (mAbs) in reducing vascular hyperpermeability and increasing the cytolysis of DENV-infected cells [25].

Another promising target for neutralizing antibodies is the domain III of the DENV envelope protein (EDIII). In a humanized mouse model, humanized mAb 513 targeting the EDIII neutralized all four dengue serotypes without antibody-mediated enhancements [26]. In another new development, the recombinant monoclonal antibody (VIS513) developed in India was recently tested in its first-in-human study in healthy adults in Australia and was found to be safe and well-tolerated, thus supporting further clinical development [27].

## 8. Dengue Therapeutics and Antivirals

Several targets of direct-acting antivirals (DAAs) and host-directed antivirals (HDAs) have been under investigation over the past decade. However, this journey is marred with challenges to develop compounds with high efficacy, safety, and stability, but with a low risk of resistance. The DAA candidates that have been most extensively evaluated to date target the DENV structural protein (E protein) and nonstructural (NS5 and NS3) proteins. Since DAAs target viral proteins, they generally offer lower toxicity and a wider treatment window but show a relatively higher risk of resistance. On the other hand, host-directed antivirals (HDAs) interfere with the host cellular pathways and pose a lower risk of resistance, but have a lower efficacy window. Cellular targets such as α-glucosidase and inosine monophosphate dehydrogenase, among several others, are currently undergoing further research [28]. JNJ-1802, a pan-serotype dengue antiviral small molecule, is the first antiviral to demonstrate safety and tolerability in a first-in-human study, making it a promising candidate for further clinical development [29]. Other dengue therapeutic candidates, such as metformin, rupatadine, and montelukast, among others, are in further stages of clinical testing.

## 9. The Dengue Alliance

One of the major drawbacks of current dengue management is the focus on a reactive strategy as opposed to an active preventive strategy to prevent progression to severe dengue (SD). There is an urgent need for safe, effective, affordable, and accessible antiviral therapies to prevent the progression to SD. To meet this demand, the endemic countries must carry out the research and development of dengue therapeutics together.

One such initiative is the Dengue Alliance, launched in 2022, which is a global partnership led by institutions from dengue-endemic countries. Participating institutions include the Translational Health Science and Technology Institute, India; the Ministry of Health, Malaysia; the Faculty of Medicine at Siriraj Hospital, Mahidol University, Thailand; the Oswaldo Cruz Foundation, Brazil; and the Federal University of Minas Gerais, Brazil. The Dengue Alliance aims to develop new treatments for dengue by using repurposed drugs and combinations, along with novel antivirals, within the next five years. The Dengue Alliance is co-owned and co-funded by dengue-endemic countries, thus facilitating the sharing of knowledge, capabilities, experience, and resources [30].

## 10. Engaging Youth in Dengue Control

The Young ADVA program, with the objective of engaging youth in dengue control, was launched during the 6^th^ Asia Dengue Summit in 2023. In association with Junior Achievement (JA), one of the largest global non-profit organizations empowering students [31], ADVA launched a community participation and educational project titled ADVA/JA Dengue Slayers Challenge. This inaugural initiative aimed to develop innovative solutions for dengue control and was held across five countries (Indonesia, Malaysia, Philippines, Singapore, and Thailand), involving 459 students aged 16–19 years. The participants undertook preparatory workshops on Understanding Dengue (Dengue 101), Design Thinking, Pitching, and Presentation. Following preliminary rounds, the final competition was held during the 7^th^ Asia Dengue Summit. The ADVA/JA Dengue Slayers Challenge gathered huge print and social media coverage, reaching a vast audience of youth, policymakers, and healthcare providers and providing a stepping stone for similar community engagement initiatives in the future.

## 11. Conclusions

It is imperative to translate dengue research from the bench to the bedside to directly improve patient outcomes and achieve the target of zero dengue deaths. An integrated, holistic, and coordinated effort through multisectoral collaboration between clinicians, researchers, academia, industry, and community is crucial to bring the research from theory to reality. We need to use all the possible tools in the dengue control armamentarium from innovative vector control to vaccines, antivirals, and host-directed therapies. However, tools are of limited value if they are not implemented appropriately. Strong leadership, effective policies, political will and commitment, and trust between researchers, policymakers, and the community are key for the successful implementation of dengue control strategies.

## Data Availability

Not applicable.

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
