# Peer review of "Proceedings of the 7th Asia Dengue Summit, June 2024"

_vaccines, 2025, doi:10.3390/vaccines13050493_

Round 1
Reviewer 1 Report
Comments and Suggestions for Authors
The conference report by Zulkifli I et al provides an overview of the different topics addressed during the 7th ADS meeting that was held in Malaysia in June 2024.
It is a pity that there is no mention of the need for Dengue genomic surveillance programs. Was this topic not discussed at this conference?
Lines 82-85: Although this is a conference report and the % reported by the authors represent the most relevant information for the reader, I would suggest including the number of cases for each group to demonstrate the robustness of the % reported.
Line 82: To avoid superlatives, please remove the term “exceptional”
Line 119: Please specify the geographical distribution of these 17 countries (at least by continent).
Line 181: A brief sentence explaining the absence of reliable serological tests (apart seroneutralization assays) due to a lack of specificity (cross-reactions between flaviviruses) would be welcome.
Author Response
Comment 1: It is a pity that there is no mention of the need for Dengue genomic surveillance programs. Was this topic not discussed at this conference?
ANSWER: No, this was not in the conference agenda.
Comment 2: Lines 82-85: Although this is a conference report and the % reported by the authors represent the most relevant information for the reader, I would suggest including the number of cases for each group to demonstrate the robustness of the % reported.
ANSWER: Inserted as suggested
Comment 3: Line 82: To avoid superlatives, please remove the term “exceptional”
ANSWER: Removed as suggested
Comment 4: Line 119: Please specify the geographical distribution of these 17 countries (at least by continent).
ANSWER: Inserted as suggested
Comment 5: Line 181: A brief sentence explaining the absence of reliable serological tests (apart seroneutralization assays) due to a lack of specificity (cross-reactions between flaviviruses) would be welcome.
ANSWER: Did not include as this takes the discussion into a completely different direction
Reviewer 2 Report
Comments and Suggestions for Authors
This conference report focuses on dengue prevention and control. It has a clear overall structure and rich content, covering multiple key areas of dengue research and prevention, and provides valuable information for promoting the development of this field.
Author Response
Thank you for your review comments
Reviewer 3 Report
Comments and Suggestions for Authors
The core content of this paper is the 7th Asia Dengue Summit, themed "Road Map to Zero Dengue Deaths," emphasized integrated strategies combining innovation and collaboration to combat dengue. Key advancements included the Wolbachia method, which reduced dengue incidence by 77% in Indonesia and 62.4% in Malaysia, with WHO endorsement for broader implementation. Achieving zero deaths demands multisectoral efforts, combining vaccines, vector control, therapeutics, and policy commitment, underpinned by global collaboration. Overall, these conclusions may be beneficial to communities, but they need major revision before acceptance:
- Line27, Aedes should be in italic format, it is recommended to check the Latin name format for the full paper.
- Line43, ";" is used between keywords, not ",".
- Line70, the title format is incorrect, it is recommended to check the title in full paper.
- Line134, one more space, it is recommended to check the full paper.
- The title of point 3 and point 10 is the same, if the content is different, it is recommended to distinguish.
- It is recommended to categorise the following contents, such as predictive tools (early warning tools, infection models, potential biomarkers), current prevention and control strategies (WMP, AWED), as well as the research progress of treatments (vaccines, antibodies, antiviral drugs), the significance of the meeting (last 9 and 10 points), and the problems and challenges faced.
- Line112, 123,133, the references should be cited on the left of ".", the format is incorrect, it is recommended to check the full paper.
- Many of the formats of the references are incorrect, such as 6and 11, line 294, 308, some names are full names, some abbreviations, it is recommended to check the full
- Some references have doi numbers and some do not, it is recommended to check the full paper.
Author Response
Comment 1: Line27, Aedes should be in italic format, it is recommended to check the Latin name format for the full paper.
ANSWER: Changed and checked as suggested
Comment 2: Line43, ";" is used between keywords, not ",".
ANSWER: Changed as suggested
Comment 3: Line70, the title format is incorrect, it is recommended to check the title in full paper.
ANSWER: Changed and checked as suggested
Comment 4: Line134, one more space, it is recommended to check the full paper.
ANSWER: Changed and checked as suggested
Comment 5: The title of point 3 and point 10 is the same, if the content is different, it is recommended to distinguish.
ANSWER: Amended point 10 title
Comment 6: It is recommended to categorise the following contents, such as predictive tools (early warning tools, infection models, potential biomarkers), current prevention and control strategies (WMP, AWED), as well as the research progress of treatments (vaccines, antibodies, antiviral drugs), the significance of the meeting (last 9 and 10 points), and the problems and challenges faced.
ANSWER: The section titles already reflect these classification, and the last point has been edited to a conclusion.
Comment 7: Line112, 123,133, the references should be cited on the left of ".", the format is incorrect, it is recommended to check the full paper.
ANSWER: In papers published by MDPI journals, references are cited in brackets after the period or other punctuation at the end of the sentence.
Comment 8: Many of the formats of the references are incorrect, such as 6and 11, line 294, 308, some names are full names, some abbreviations, it is recommended to check the full
ANSWER: Checked and amended as recommended
Comment 9: Some references have doi numbers and some do not, it is recommended to check the full paper.
ANSWER: Checked and updated, however do note that dome references are either not published in journals, or not indexed in DOI. Others are online publications from the respective bodies.
Round 2
Reviewer 3 Report
Comments and Suggestions for Authors
The author of this article has made detailed revisions to the previously proposed revision suggestions and has met the publication standards.